# Secure privacy-preserving record linkage system from re-identification attack

**Sejong Lee**[1], **Yushin Kim**[1], **Yongseok Kwon**[1], **Sunghyun Cho**[2]*

**1** Department of Computer Science and Engineering, Major in Bio-Artificial Intelligence, Hanyang University, Ansan-si, Gyeonggi-do, South Korea, **2** Department of Computer Science and Engineering at Hanyang University ERICA, Ansan-si, Gyeonggi-do, South Korea

* chopro@hanyang.ac.kr

**Data Availability Statement:** The simulation files and data relevant to this study are from Anushka Vidanage, Peter Christen, Thilina Ranbaduge, and Rainer Schnell. 2020. 'A Graph Matching Attack on Privacy-Preserving Record Linkage.' In Proceedings of the 29th ACM International

## Abstract

Privacy-preserving record linkage (PPRL) technology, crucial for linking records across datasets while maintaining privacy, is susceptible to graph-based re-identification attacks. These attacks compromise privacy and pose significant risks, such as identity theft and financial fraud. This study proposes a zero-relationship encoding scheme that minimizes the linkage between source and encoded records to enhance PPRL systems' resistance to re-identification attacks. Our method's efficacy was validated through simulations on the Titanic and North Carolina Voter Records (NCVR) datasets, demonstrating a substantial reduction in re-identification rates. Security analysis confirms that our zero-relationship encoding effectively preserves privacy against graph-based re-identification threats, improving PPRL technology's security.

## 1. Introduction

The escalating collection of user data for personalized services across various sectors in media advertising, the Internet of things, and healthcare has heightened privacy concerns significantly [1–6]. Integrating data from diverse sources through record linkage exacerbates these privacy challenges [7, 8], especially when personally identifiable information (PII) like name, email address, or passport number is employed for linkage [9–11]. Despite its high accuracy, this conventional approach poses substantial security risks by potentially exposing individuals to identity theft and misuse of information [12–14].

In response to these vulnerabilities, Privacy-Preserving Record Linkage (PPRL) technologies have been developed to link records across different datasets while privacy preservation securely [15–18]. However, PPRL's effectiveness is undermined by sophisticated graph-based re-identification attacks [19–21]. These attacks can decrypt anonymized records, thus nullifying PPRL's privacy-preserving capabilities [22]. The gravity of these attacks cannot be overstated, as they can lead to severe consequences, including identity theft and financial fraud. Fig 1 presents a simplified overview of the graph-based re-identification attack proposed in [22].

Although innovative, existing solutions, such as split bloom filters and two-step hash methods, fall short of fully safeguarding against these re-identification attacks [23–26]. These

Conference on Information & Knowledge Management (CIKM '20). Association for Computing Machinery, New York, NY, USA, 1485–1494. https://doi.org/10.1145/3340531.3411931. The authors confirm that others would be able to access these data in the same manner as them and that the authors did not have any special access privileges that others would not have.

**Funding:** This work was supported by Korea Research Institute for defense Technology planning and advancement(KRIT) grant funded by the Korea government(DAPA(Defense Acquisition Program Administration)) (No. KRIT-CT-22-021, Space Signal Intelligence Research Laboratory, 2022). The funder had no role in study design, data collection and analysis, publication decisions, or manuscript preparation.

**Competing interests:** The authors have declared that no competing interests exist.

methods do not adequately address the inherent relationship between source and anonymized records, leaving a window for potential privacy breaches [27].

This study proposes a novel zero-relationship encoding scheme to fortify PPRL's security against graph-based re-identification attacks. Our method significantly diminishes the risk of privacy breaches by effectively minimizing the relationship between source and encoded records. We empirically validate the efficacy of our approach through simulations using the Titanic and North Carolina Voter Records (NCVR) datasets, demonstrating superior resistance to re-identification attacks compared to existing methodologies. The main contributions of this study are as follows:

- We propose a zero-relationship encoding scheme to defend against re-identification attacks within PPRL systems. Unlike traditional anonymization techniques, our method ensures minimal relational links between the source and encoded records, significantly enhancing privacy preservation.

- We propose a novel PPRL framework incorporating the zero-relationship encoding method. This framework is tailored to resist graph-based re-identification attacks, addressing a critical security vulnerability in existing PPRL technologies.

- Through comprehensive simulations using real-world datasets (Titanic and North Carolina Voter Records), we demonstrate the superior resistance of our proposed method against re-identification attacks. The proposed methods significantly reduce re-identification rates compared to existing techniques.

- We introduce a concrete method for evaluating the effectiveness of different encoding methods in preserving privacy and resisting re-identification attacks. We utilize entropy and information gain as metrics to assess the security performance of the encoding method quantitatively. This contribution provides a practical tool for researchers and practitioners to evaluate the security efficacy of their PPRL solutions.

Our research fills a security vulnerability in PPRL by providing a robust solution to the re-identification attacks. It underscores the necessity for encoding methods that consider the relationship between source and anonymized records, offering a blueprint for future developments in secure PPRL frameworks.

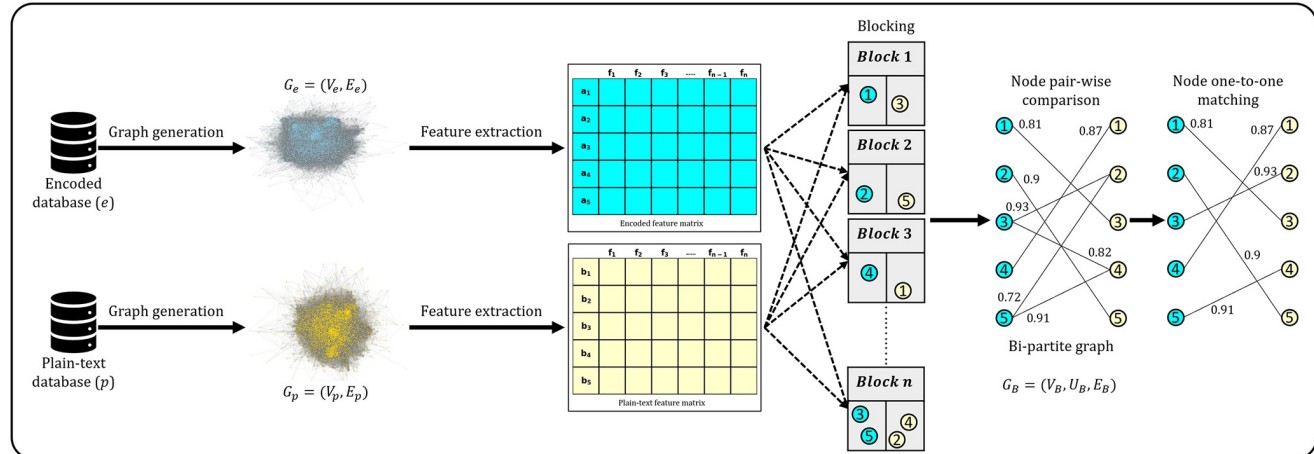

**Fig 1. Overview of the graph-based re-identification attack [22].**

The remainder of this paper is organized as follows: The privacy-preservation study to improve PPRL's security section introduces related studies. The Methods section details the proposed zero-relationship encoding method and demonstrates that the PPRL system is safe from re-identification attacks. We evaluate the proposed method's performance in the Results section and compare it with related studies. Finally, the Conclusion section concludes the paper.

## 2. Privacy-preservation study to improve PPRL's security

In recent years, numerous security methods have been developed to address the security vulnerability in PPRL. These security methods are classified into five categories: cryptography [28, 29], embedding techniques [30], differential privacy [31, 32], statistical linkage key [33, 34], and probabilistic methods [24, 35–37]. Among these, probabilistic methods have garnered attention because of their computational efficiency and capability to manage large-scale datasets [38]. However, security methods based on probabilistic approaches are vulnerable to privacy attacks, such as re-identification, inference, man-in-the-middle, and dictionary attacks [21]. Privacy leaks due to privacy attacks are a severe problem that undermines the purpose of PPRL [25–27]. Therefore, various methods have been proposed to overcome these limitations.

While this study primarily focuses on enhancing the security of PPRL systems against re-identification attacks by minimizing the relationship between source and encoded records, it does not address the handling of encrypted data queries in distributed environments such as cloud computing. This aspect has been comprehensively explored in another study by the author, where an efficient privacy-preserving spatial range query for outsourced encrypted data was proposed. This study provides additional insights into managing and querying encrypted data in distributed environments, complementing the current research by focusing on different aspects of data security [39]. We introduce some studies that are representative of improving the security performance of PPRL. Table 1 summarizes the contents of related works to strengthen PPRL security.

- Ranbaduge *et al.* proposed a windowing-based XORing (WXOR) method to mitigate the vulnerability of the bloom filter encoding method [36]. The WXOR method employs a sliding window approach on the bloom filter encoding (BF), searching for adjacent bit patterns, and subsequently XORs these patterns to generate a new one. The flexibility of adjusting the window size allows optimization in terms of encoding process efficiency or security enhancement. Another study introduced a new hash-based encoding method [24]. The proposed secure two-step hash encoding method hashes private record information into a bit matrix, with each column hashed to generate an integer set representation of a sensitive input value. Simulations using real data demonstrated that the method can substantially improve record linkage quality in large databases. However, both encoding methods are still vulnerable to re-identification attacks. Both approaches retain the characteristics of the source records in the encoded records, allowing the re-identification of records using features extracted through separate processing [40, 41].

- Smith proposed the tabulation min-hash (TMH) encoding method as a remedy to the vulnerabilities in the bloom filter encoding method [37]. The primary objective of this method is to safeguard privacy by circumventing the simple mapping of plaintexts encoded into q-grams. TMH utilizes a bit array of length $l$ in tandem with a set of lookup tables to encode the records. Each set of lookup tables consists of multiple tables and keys. These keys, which symbolize random bit strings, are derived from the MinHash values of the q-gram set. These random bit strings undergo an XOR operation, resulting in a definitive bit string aggregated

**Table 1. Related work to improve the security of PPRL.**

| Year | Author | Description | Entropy | Identification risk | Limitation |
|---|---|---|---|---|---|
| 2017 | Smith D. | • Proposed tabulation min-hash to address vulnerabilities in BF encoding, enhancing privacy by preventing simple plaintext-to-q-gram mappings.<br>• Encodes records using min-hash values and a bit array to improve security. | Medium | Medium | Maintaining essential bits for similarity in encoding. |
| 2020 | Ranbaduge et al. | • Introduced a windowing-based XOR method to mitigate BF encoding vulnerabilities.<br>• Utilizes sliding windows to combine bit patterns within q-grams using XOR, with adjustable window sizes to optimize security and efficiency. | Medium | Low | Relationships between source and encoded records. |
| 2020 | Ranbaduge et al. | • Presented a two-step hash encoding method using a bit matrix for secure record linkage.<br>• Demonstrated improved linkage quality through simulations on large databases. | Medium | Low | Relationships between source and encoded records. |
| 2021 | Nóbrega et al. | • Proposed a blockchain-based PPRL approach to counter unrealistic adversarial models.<br>• Utilizes splitting bloom filters to reduce data exchange, minimizing security risks during record linkage. | Medium | Medium | Re-identification risks if a user becomes malicious during BF exchange. |
| 2022 | Victor C. et al. | • Proposed an autoencoder-based method to transform BFs into numerical vectors, enhancing security by preventing known BF attacks.<br>• Ensures data privacy and comparability through a mapping function for BF encoding and decoding. | High | Low | Autoencoder sensitivity. |
| 2023 | Siyu Y. et al. | • Proposed a PPRL method using a Siamese neural network (SNN) for secure and accurate record matching.<br>• Combines flat and transform bloom filters to handle diverse data types and improve performance in error-prone scenarios. | High | Very Low | Scalability considerations and adaptation to data errors. |
| 2023 | Frederik A. et al. | • Proposed a BF with diffusion (BFD) scheme, extending conventional BFs with a linear diffusion layer.<br>• Enhances security against known attacks while preserving or improving linkage quality. | High | Low | Complexity in balancing cost and fairness. |
| 2024 | Ours | • Proposes a zero-relationship encoding method to prevent re-identification attacks and preserve privacy.<br>• Applies XOR-folding to hash values for generating encoded records, enhancing security and efficiency. | Very High | Very Low | Processing of records with a small number of attributes. |

to form the final bit vector. Although TMH offers strong security performance owing to its high computational complexity, it remains vulnerable to re-identification attacks. This vulnerability stems from the approach of preserving the minimum number of bits for similarity comparison during encoding. These minimum bits retain the characteristics of the source record, enabling attackers to extract features from the encoded record and subsequently re-identify it [42]. Therefore, TMH is unsafe against re-identification attacks.

• Nóbrega *et al.* highlighted the shortcomings of existing PPRL research that assumed an unrealistic adversarial model and proposed an auditable blockchain-based PPRL (ABEL) model that considers a more realistic adversarial model [23]. Additionally, they proposed a splitting bloom filter (SBF) method to address the vulnerabilities inherent in the conventional bloom filter. This method minimizes the information shared between parties involved in the record linkage process. Unlike traditional methods that require the exchange of the entire bloom filter to check for similarity during the comparison stage of PPRL, the SBF method only

exchanges portions of the original bloom filter. The method reduces the risk of data leakage by limiting the amount of shared information, thereby enhancing security. However, ABEL still presents security vulnerabilities. In the bloom filter segment exchange protocol proposed for database owners, a re-identification attack becomes feasible if a user receiving a bloom filter segment from another user turns malicious [43, 44]. An attacker possessing the same encoding knowledge as a regular user can utilize their database to find a set of q-grams that can be hashed to that specific location in the segment, resulting in privacy breaches through the re-identification of records encoded by regular users.

Existing studies have often overlooked the relationship between source data and encoded records during the encoding process. The closer this relationship, the easier it is for an attacker to decrypt the encoded data and infer the source data. This relationship is closely linked to the entropy and re-identification risk of the encoded data.

Entropy measures the randomness of the encoded data. Higher entropy indicates that the data is more unpredictable, making it more secure against attacks. Conversely, lower entropy means that the encoded data is more structured and predictable, increasing the likelihood of an attacker inferring the source data. This is particularly problematic when the relationship between the source data and encoded data is strong, as low entropy can make this inference easier.

Re-identification risk refers to the likelihood that a source record can be re-identified from the encoded data. A higher re-identification risk suggests that the encoded data is more likely to be linked back to the source data, posing a significant privacy threat. Recent studies have demonstrated that if an attacker knows the encoding parameters, they can identify a source record by launching a dictionary attack on the encoded record [20, 22].

Therefore, it is crucial to develop a security method that maximizes entropy and minimizes re-identification risk during the encoding process, while also considering the relationship between source and encoded records. This approach ensures a balance between the randomness of the encoded data and the protection of the source data's privacy, making it harder for attackers to exploit any potential vulnerabilities.

## 3. Methods

This section describes the proposed zero-relationship encoding method designed to prevent re-identification attacks. We use the concepts of PPRL and anonymization to formulate the privacy issues posed by the re-identification attacks addressed in this study [23]. Especially, we propose the zero-relationship encoding algorithm, which eliminates the relationship between the source and the encoded records to prevent re-identification attacks.

### 3.1. Problem formulation

Addressing the privacy preservation issue in the PPRL system, we assume that all users participating in the system are semi-trusted. Partial trust is accorded to these participants, as they could potentially be honest-but-curious or hidden malicious adversaries [45, 46]. All threats are assumed to be logical ones carried over the network rather than physical ones.

Table 2 shows the notations used throughout this paper. In the PPRL system, multiple users $P$ participate. Each user, denoted as $p$, owns an individual database $DB_p$. This database consists of a set of entities $e_n = attr_1, \ldots, attr_i$, expressed as $DB_p = e_1, \ldots, e_n$. PPRL anonymizes the entity of each user to preserve privacy as denoted by $e_n^\tau = anonymize(e_n)$. Participants perform the record linkage process using an anonymized database $DB_p^\tau = Encode(e) | \forall e \in DB_p$. A

**Table 2. Notations.**

| Symbol | Description |
|---|---|
| $DB_p$ | Database of participant |
| $DB_p^\tau$ | Anonymized database |
| $P$ | Participant in PPRL |
| $attr$ | Attribute of an entity |
| $k$ | Number of attributes |
| $\mu$ | Hash seed |
| $H$ | List of hash functions |
| $e_n$ | Entity |
| $e^\gamma$ | Entity with selected attributes |
| $\phi$ | Min-hash XOR set |
| $e^\tau$ | Anonymized entity |
| $Zero(DB_p, \mu, s)$ | Zero-relationship encoding method |
| $Unique(DB_p)$ | Number of unique values in the attributes |
| $Total(DB_p)$ | Total number of values in the attributes |
| $Card(attr_i)$ | Degree of duplication for attributes |
| $\delta$ | Security rank of the attributes |
| $s$ | Number of attributes for selection |
| $\gamma$ | Selected attribute set |
| $attr_i'$ | hash value of the attribute |
| $\zeta$ | Hashed attributes set in $\gamma$ |
| $\theta$ | Combination of elements in $\zeta$ |
| $\Xi(\cdot)$ | Record linkage decision model |

decision model $\Xi(\cdot)$, which is a critical component of PPRL, identifies and links entities referring to the same subject.

$$\Xi(e_i^\tau, e_j^\tau) | \{e_i^\tau, e_j^\tau\} \in DB_i^\tau \times DB_j^\tau \tag{1}$$

The decision model compares and classifies all anonymized entities between the databases intended for linkage. The comparison process computes the similarity between entities using the Jaccard similarity measure and classifies them into matches (M) or mismatches (U) based on the computed similarity value.

$$\Xi(e_1^\tau, ... e_i^\tau) = \begin{cases} M \Leftrightarrow \forall(e_j^\tau, e_{j+1}^\tau) : (e_1^\tau, ... e_i^\tau), e_j^\tau = e_{j+1}^\tau \\ U \Leftrightarrow \exists(e_j^\tau, e_{j+1}^\tau) : (e_1^\tau, ... e_i^\tau), e_j^\tau \neq e_{j+1}^\tau \end{cases} \tag{2}$$

where $(e_j^\tau, e_{j+1}^\tau)$ is anonymized entity set.

A re-identification attack, defined as an attacker's attempt to deduce the source record from an anonymized record without a permit, poses a significant threat to the security of a PPRL system. Attackers execute the record-matching process using datasets for re-identification attacks. They aim to identify records that bear a similar anonymized form to the anonymized record. To increase the success rate of their re-identification efforts, attackers seek

records with a high probability of matching. This challenge is formulated as follows:

$$\underset{e_n \in DB_p}{\mathrm{argmax}} \; \Pr\left(\Xi(\mathrm{Encode}(e_n), e^\tau) = M\right) \qquad (3)$$

Eq 3 delineates the process within a PPRL system of identifying a record within the database $DB_p$ that maximizes the probability of a match with a given anonymized record $e^\tau$. This models the attacker's scenario of attempting re-identification to locate the source record. The expression $\Pr(\Xi(\mathrm{Encode}(e_n), e^\tau) = M)$ signifies the probability of deriving $e^\tau$ from $e_n$. $\Xi$ is the decision model previously defined, tasked with determining whether a pair of anonymized records refer to the same entity. However, the inference of $e_n$ from $e^\tau$ presents considerable difficulty, with the attacker's objective being to identify the $e_n$ that maximizes the probability of a match. This entails an exhaustive consideration of all potential $e_n$s to select the record with the highest probability of correlation.

The paramount goal of this research is to preserve privacy breaches stemming from re-identification attacks. Thus, we strive to diminish the success rate of such attacks by employing an encoding method impervious to re-identification threats. This principal objective is mathematically formulated as follows:

$$\underset{\mathrm{Encode}}{\mathrm{argmin}} \; \underset{e_n \in DB_p}{\mathrm{argmax}} \; \Pr\left(\Xi(\mathrm{Encode}(e_n), e^\tau) = M\right) \qquad (4)$$

Eq 4 seeks an encoding method that minimizes the probability established in Eq 3. The argmax component represents the attacker's endeavor to pinpoint the $e_n$ that ensures the maximum probability of a match. Conversely, the argmin aspect indicates the quest for an encoding method that minimizes this match probability. Eq 4 accentuates how the encoding method can bolster resistance to re-identification attacks. An efficacious encoding method minimizes the linkage between $e_n$ and $e^\tau$, thereby reducing the likelihood of an attacker successfully deducing the source record. Consequently, Eq 4 is instrumental in determining encoding methods adept at countering re-identification attacks.

## 3.2. Zero-relationship encoding method

The proposed zero-relationship encoding method eliminates the relationship between the source and encoded records. An anonymization process that leverages the various attributes of the source record produces an encoded outcome with minimal ties to the original form. Our approach prevents an attacker from obtaining specific information from the encoded record to re-identify it. The procedure for the zero-relationship encoding method is shown in Fig 2. As outlined in Algorithm 1, this method calculates the cardinality of the attributes in the user's database. Cardinality, an indicator of the degree of duplication across attributes, aids in determining the security priority of each attribute [47]. Attributes with more duplications have lower security ranks, whereas those with fewer have higher security ranks. This is because attributes with unique values are more effective in identifying specific entities, thus posing a risk to privacy if disclosed. Conversely, attributes with high cardinality can compromise distinguishability. To balance security and utility, we excluded attributes with extraordinarily high and low cardinality values from the encoding process.

The cardinality of each attribute, denoted as $Card(attr_i)$, is calculated as $1 - \frac{|Unique(DB_p)|}{|Total(DB_p)|}$, where $Unique(DB_p)$ and $Total(DB_p)$ represent the number of unique values for attribute $attr_i$ in database $DB_p$ and the total number of values for the same attribute in the database, respectively. After computing the cardinalities, we generate a security rank ($\delta$) for each attribute and sort the attributes accordingly. We select $s$ attributes from these sorted attributes, excluding

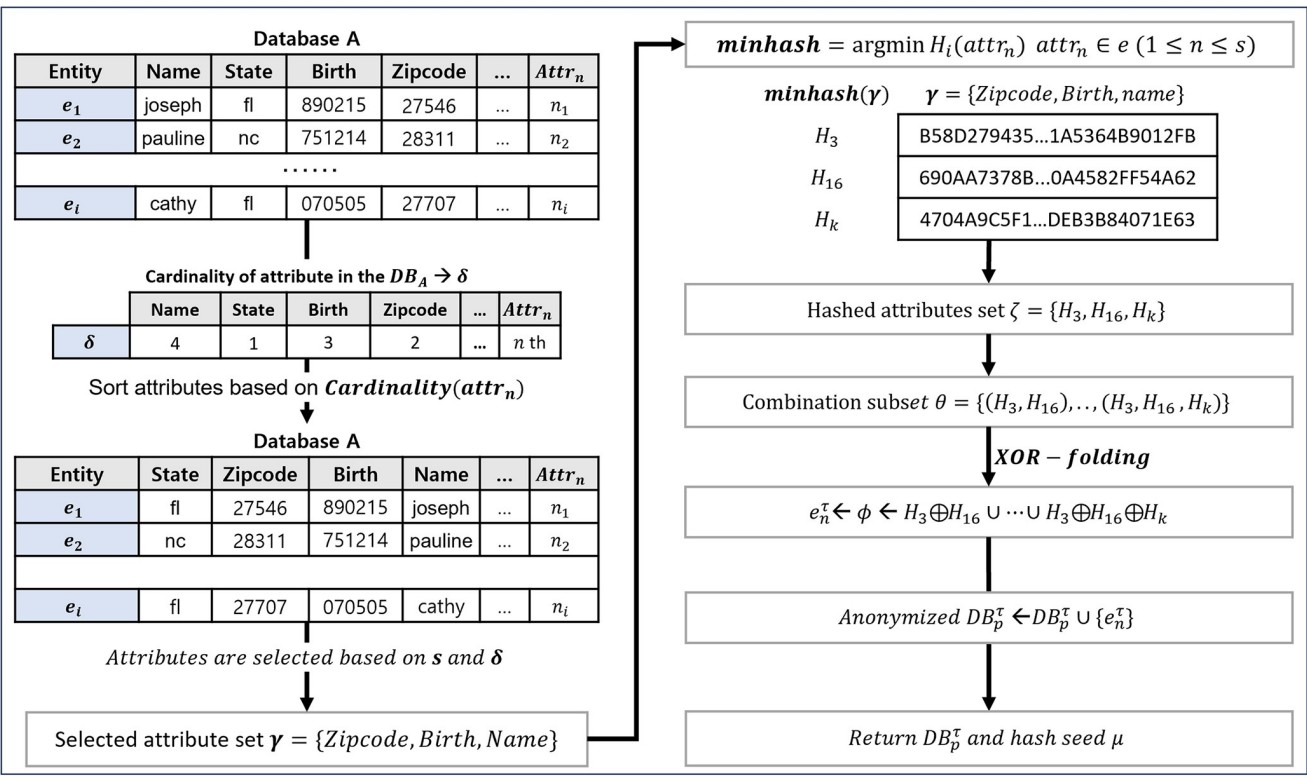

**Fig 2. The process of the proposed zero-relationship encoding method.**

those with the highest and lowest ranks. This step ensures a balance between data security and the preservation of adequate identifiable information. The min-hash function, given by $attr'_i \leftarrow \text{argmin}_{h \in H} h(attr_i)$, is employed to obtain the smallest hash value for each selected attribute from the set *H*. By employing this hashing method, each selected attribute is transformed into its respective hash value, enhancing the security of the dataset. This variation in the data poses a challenge for adversaries, complicating their efforts to discern the original information. The next step involves combining the hashed attributes using XOR operations to generate anonymized attributes. XOR operations are executed for all subsets of the hashed attributes, and the encoded results are added to the anonymized dataset $DB_p^\tau$. Using XOR, an operator representing the difference between two binary numbers, minimizes the relationship between the source and the encoded data. Because the hashing process obfuscates the encoded data, they demonstrate strong resilience against re-identification attacks based on security analysis. Even if an attacker attempts to re-identify using features extracted from the encoded data in conjunction with their information, they will only obtain meaningless values due to the severed relationship between the encoded data and the original text. The resulting anonymized database and hash seed $\mu$ are returned as outputs of the function.

**Algorithm 1** Proposed zero-relationship encoding method

```
Input: Participant's Database DBₚ, List of the hash function H, Number
       of Selected Attributes s
Output: Anonymous Database DBₚᵗ, Hash Seed μ
 1: DBₚᵗ ← {}
 2: k ← Number of attributes in DBₚ
 3: for each entity eₙ in DBₚ do
```

```
4:    Compute Card(attr_i) = 1 - |Unique(DB_p)|/|Total(DB_p)|
      for each attribute attr_i in e_n
5:    Sort the attributes based on δ
6:    γ ← Select s attributes from δ
      excluding the highest and lowest rank
7:    ζ ← {}
8:    for each attr_i in γ do
9:       attr'_i ← argmin_{h∈H} h(attr_i)
10:      ζ ← ζ ∪ {attr'_i}
11:   end for
12:   Initialize φ ← {}
13:   for each subset θ of ζ do
14:      φ ← φ ∪ {⊕_{attr'_i∈θ} attr'_i}
15:   end for
16:   e_n^τ ← φ
17:   DB_p^τ ← DB_p^τ ∪ {e_n^τ}
18: end for
19: return DB_p^τ, μ
```

The proposed method minimizes the possibility of re-identifying $e^\tau$ as $e$ by eliminating any explicit relationships between records. This approach is rooted in randomized hash functions in conjunction with XOR operations, resulting in multiple plausible fictitious source records for each encoded record. Fictitious source records introduce inherent uncertainty when attempting to identify and classify $e^\tau$ during a re-identification attack, thereby impairing the efficacy of the attack model. Hence, the potential for associating $e^\tau$ with its original $e$ is significantly diminished, which aligns with the objective articulated in Eq (4).

The security of the proposed method against re-identification attacks can be theoretically demonstrated theoretically. The success probability of a re-identification attack can be calculated based on the number of bits required to identify $e^\tau$. In an encoded record of length $l$, the likelihood of an attacker accurately guessing a randomly chosen bit is 1/2, and the probability of correctly guessing all $l$ bits is $\left(\frac{1}{2}\right)^l$.

However, the probability of an attacker identifying $e^\tau$ is $\frac{1}{n+1}$ because the proposed encoding method generates multiple plausible fictitious source records. Thus, the success rate of a re-identification attack on a record encoded using the proposed method is represented as

$$P_{\text{attack}} = \frac{1}{n+1} \times \left(\frac{1}{2}\right)^l \tag{5}$$

Eq (5) consists of two primary components. The first component, $\frac{1}{n+1}$, represents the probability that one of the $n$ plausible fictitious source records produced by the encoding method corresponds to the actual record. The second component, $\left(\frac{1}{2}\right)^l$, is the probability of accurately guessing a randomly chosen bit.

Our goal is to ensure that the proposed method renders $P_{\text{attack}}$ a negligible function $\epsilon(l)$. The function $\epsilon(l)$ denotes the negligible function of a security parameter commonly employed in cryptography. The term "negligible" indicates that a specific encryption technique or method is considered secure from external attackers. The value of the $\epsilon(l)$ function converges to zero when $l$ increases. The closer the value of $\epsilon(l)$ is to 0, the less likely the attack is to succeed, indicating security against threats.

Assuming that the length $l$ of the encoded information is sufficiently large, hash algorithms such as SHA-256 and $P_{\text{attack}}$ can be represented as $\epsilon(l)$. Consequently, the proposed method provides sufficient anonymity for $DB_p^\tau$ to protect privacy from re-identification attacks.

## 4. Results: Evaluation and discussion

### 4.1. Simulation setup

The performance of the proposed encoding method was assessed through experiments utilizing a re-identification attack simulator provided by [22]. The simulator, implemented in Python 2.7, was executed on a PC with a 64-bit Intel(R) Xeon(R) W-3245 3.20 GHz CPU and 256 GB of memory, running Ubuntu 20.04.1. Access to the utilized simulator is available via the link provided by the authors at https://dmm.anu.edu.au/pprlattack/.

The simulation files and data relevant to this study are from Anushka Vidanage, Peter Christen, Thilina Ranbaduge, and Rainer Schnell. 2020. "A Graph Matching Attack on Privacy-Preserving Record Linkage." In Proceedings of the 29th ACM International Conference on Information & Knowledge Management (CIKM '20). Association for Computing Machinery, New York, NY, USA, 1485–1494. https://doi.org/10.1145/3340531.3411931. The authors confirm that others would be able to access these data in the same manner as them and that the authors did not have any special access privileges that others would not have.

For the simulations, two types of public datasets were employed: the Titanic dataset [48] and the NCVR database [49]. The Titanic dataset utilized attributes such as the first and last names of 1,317 passengers. The NCVR dataset (accessed in June 2014) contains information on 224,073 North Carolina voters across 19 attribute fields. Due to numerous null values across fields, which could introduce noise or errors if used without preprocessing, a subset of 4,401 voter records without any null values was employed for accurate simulation. In the simulations based on the NCVR dataset, attributes with the highest and lowest cardinality values were excluded, and a subset of the remaining 17 attributes was randomly selected for use.

### 4.2. Performance evaluation

This section evaluates the proposed zero-relationship encoding method through a graph-based re-identification attack simulator. Our assessment focused on the method's effectiveness in preserving privacy by measuring the ratio of re-identified records. The encoding methods included in our comparison are the Bloom filter [36], Two-step hash [24], and Tabulation min-hash [37], all representing established privacy preservation techniques.

The experiments utilized two distinct datasets to conduct the evaluations: the Titanic and NCVR datasets, the results of which are illustrated in Figs 3 and 4, respectively. We observed that the zero-relationship encoding method significantly reduced the ratio of re-identified records across all matching algorithms compared to other encoding methods. The simulation results demonstrate that our encoding method effectively counters re-identification attacks by severing the explicit relationship between records.

The experiments utilizing the Titanic dataset demonstrated that traditional encoding methods like the Bloom filter, Two-step hash, and Tabulation min-hash resulted in a significant proportion of re-identified records. On average, these methods experienced re-identification rates of approximately 87%, 94.98%, and 50%, respectively, across various matching algorithms. In stark contrast, the proposed zero-relationship encoding method exhibited a remarkably lower average re-identification rate of around 1.3%, underscoring its superior performance in preserving privacy.

The NCVR dataset provided further evidence of the proposed method's robustness. The Bloom filter re-identified over 3,990 records for each matching algorithm, while the Two-step hash method showed a near-complete re-identification rate. The Tabulation min-hash offered an improved outcome yet allowed more than 1700 records to be re-identified. Strikingly, the

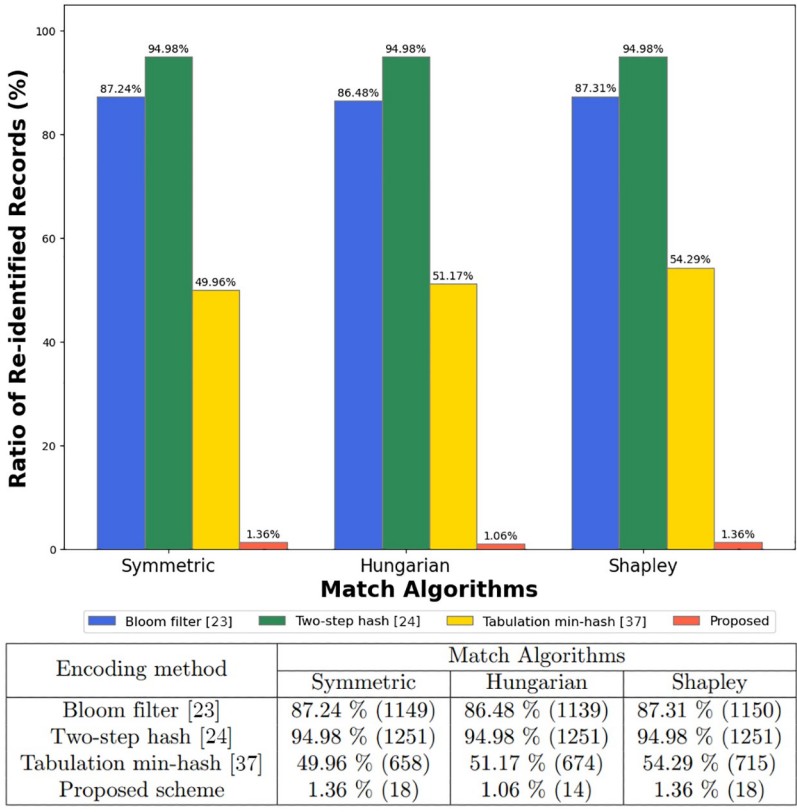

| Encoding method | Match Algorithms | | |
|---|---|---|---|
| | Symmetric | Hungarian | Shapley |
| Bloom filter [23] | 87.24 % (1149) | 86.48 % (1139) | 87.31 % (1150) |
| Two-step hash [24] | 94.98 % (1251) | 94.98 % (1251) | 94.98 % (1251) |
| Tabulation min-hash [37] | 49.96 % (658) | 51.17 % (674) | 54.29 % (715) |
| Proposed scheme | 1.36 % (18) | 1.06 % (14) | 1.36 % (18) |

**Fig 3. Re-identification rates in the Titanic dataset: Comparing encoding methods and matching algorithms.**

proposed method achieved a zero re-identification rate, demonstrating an outstanding defense against re-identification attacks.

The proposed zero-relationship encoding method eliminates any linkage between the source and the encoded records, thwarting re-identification attempts. It employs a novel anonymization process that leverages the attributes of the source record to produce an outcome with minimal ties to the original data, thus significantly enhancing privacy preservation. Our method utilizes cardinality to prioritize the security of attributes. We exclude those with extremely high or low values to balance utility and security. The min-hash function is employed to obtain the smallest hash value for selected attributes, combined using XOR operations. This process not only enhances the protection of the dataset by transforming attribute data but also denies adversaries by obscuring the original information. This approach introduces uncertainty in identifying and classifying anonymized entities, significantly impairing the efficacy of potential attacks.

Our study further examined the relationship between the number of attributes used in the encoding process and the performance of the proposed method. Fig 5 shows the results of a performance measurement experiment of the unrelated encoding method depending on the number of attributes used for encoding. The NCVR dataset, which used three attributes, exhibited a lower re-identification rate than the Titanic dataset with two attributes. The graph demonstrates that as the number of attributes increases, there is a notable impact on both encoding and feature generation times. Specifically, a substantial rise in time required is observed as we move from 9 to 14 attributes. Despite this increase in computational effort, the associated rise

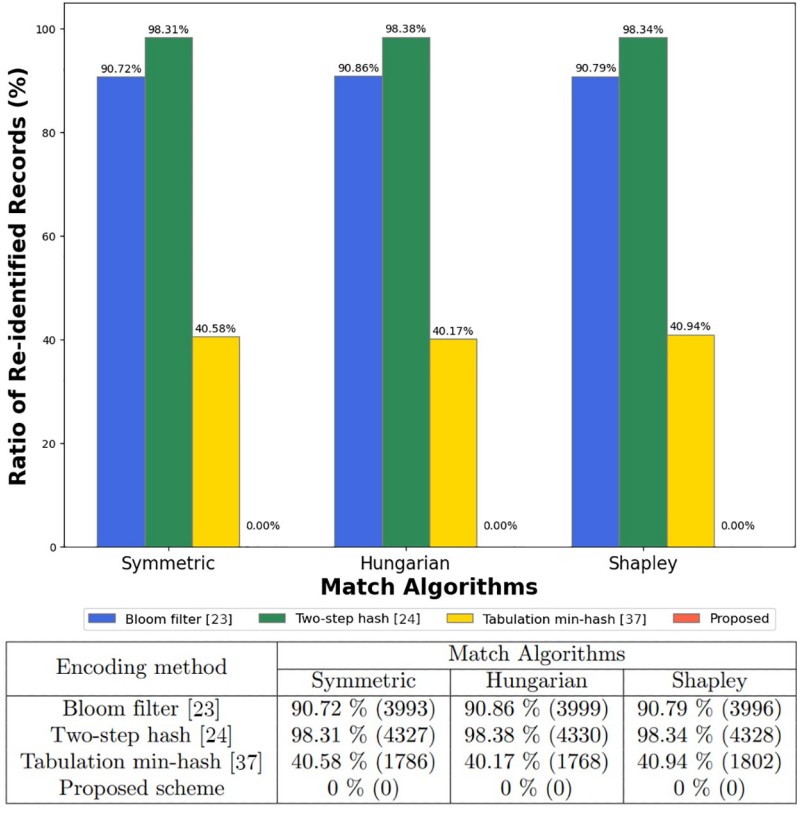

**Fig 4. Re-identification rates in the NCVR dataset: Comparing encoding methods and matching algorithms.**

in re-identification time indicates an enhancement in privacy protection. This trade-off highlights the method's scalability and potential to bolster privacy defenses with increased attribute counts, as higher complexity leads to reduced re-identification success. This nuanced understanding of performance dynamics is crucial for implementing the method in practical scenarios where attribute quantity and processing capabilities must be balanced.

The experiments conducted underscore the effectiveness of the proposed zero-relationship encoding method in mitigating the risk of data re-identification. Minimizing the relationship between the source and the encoded records provides a promising approach to privacy preservation in PPRL.

## 4.3. Security analysis

This section addresses the robustness and reliability of our proposed zero-relationship encoding method in preserving user privacy against graph-based re-identification attacks within the context of PPRL systems. Our analysis focuses on the method's capacity to obfuscate information and its defensive efficacy against potential threats.

**4.3.1. Strong privacy-preservation.** The privacy-preserving efficacy of our proposed PPRL system is substantiated through an analysis utilizing entropy, a measure reflective of randomness or uncertainty within a dataset. Entropy signifies the complexity of the anonymization applied to the data, indicating the probability of the original data being inferred by a re-identification attack [50, 51]. High entropy means a complex anonymization process,

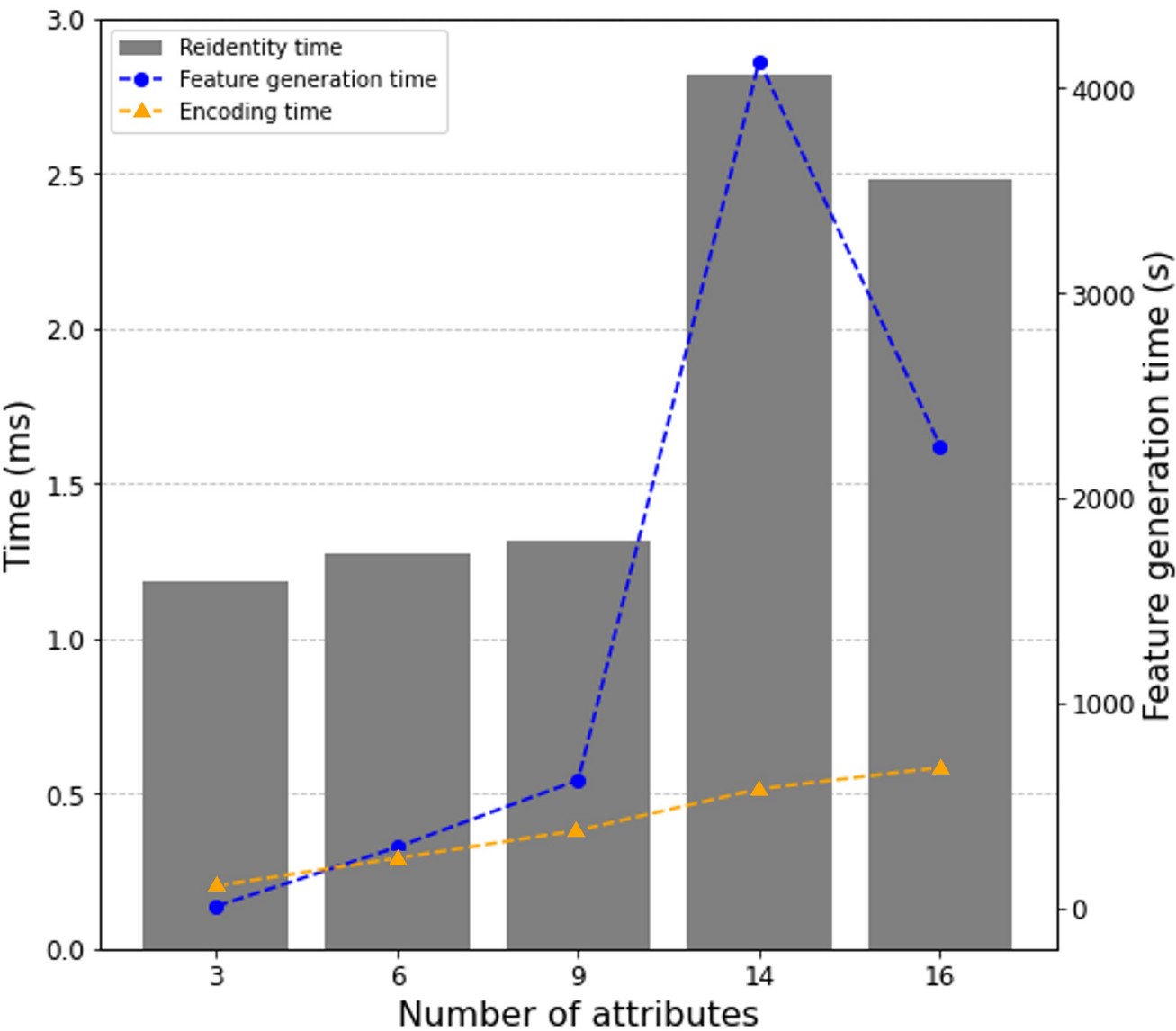

**Fig 5. Performance of the zero-relationship encoding method according to the number of attributes used for encoding.**

significantly reducing the likelihood of original data inference through re-identification attacks. This complexity confounds direct attacks and hinders advanced graph-based attacks by obfuscating the relationships within the data structure.

The entropy of a dataset is mathematically represented as:

$$H(X) = -\sum_{x \in X} p(x) \log_2 p(x) \tag{6}$$

where $H(X)$ denotes the entropy of a discrete random variable $X$, which encapsulates the encoded dataset, and $p(x)$ represents the probability of occurrence of an element $x$ in $X$.

We extend this notion to calculate the information gain ($IG$), a metric quantifying the additional information an attacker gains post-attack. This is particularly relevant in assessing the

**Table 3. Information gain.**

| Encoding method | Entropy | Match Algorithms | | |
|---|---|---|---|---|
| | | Symmetric | Hungarian | Shapley |
| Two-step hash [24] | 3.77 | 0.283 | 0.347 | 0.376 |
| Tabulation min-hash [37] | 9.965 | 0.283 | 0.347 | 0.376 |
| Proposed | 17.77 | 0.283 | 0.347 | 0.376 |

vulnerability of specific attributes within the encoding process:

$$IG(Y|X) = H(Y) - H(Y|X) \tag{7}$$

with $H(Y|X)$ being the conditional entropy of $Y$ given $X$. A lower $IG$ indicates a more resilient encoding scheme, implying that the attacker's knowledge does not significantly increase, even if certain attributes are compromised.

Our security analysis, as presented in Table 3, juxtaposes the entropy and $IG$ values calculated for both traditional encoding methods and our proposed zero-relationship encoding scheme. Notably, the proposed method achieved a high entropy of 17.77, suggesting an enhanced level of data protection against re-identification.

To confront traditional methods' limitations, we highlight the superior entropy of our method, affirming its capacity to maintain data utility while providing strong privacy. This is contrasted with traditional approaches, which often manifest lower entropy, potentially compromising privacy.

In practical scenarios, such as health data exchange or customer data protection, the proposed method's high entropy and low $IG$ values signal a significant advancement in mitigating the risks of sensitive data exposure. The method is especially pertinent for datasets with inherent attribute correlations, a common vulnerability in PPRL. Our careful selection and encoding of attributes reduce this risk, a crucial factor supported by our findings that higher frequency attributes yield greater $IG$ values.

Conclusively, this rigorous quantitative analysis underscores the zero-relationship encoding method's robustness in theoretical metrics and practical resilience against graph-based re-identification attacks, cementing its role as a pivotal advancement in PPRL privacy measures.

**4.3.2. Defense against graph-based re-identification attack.** Graphical representations of encoded datasets serve as critical indicators of an encoding method's resilience to graph-based re-identification attacks. In these representations, nodes represent unique values from a database, and edges denote similarities among these values. A graph's resilience to re-identification attacks is inversely proportional to the density of its nodes and edges: denser graphs, with many interconnected nodes, make them vulnerable to attacks as they facilitate the extraction of similarities that can lead to re-identification. Conversely, dispersed nodes could pose challenges, as unique features in such a distribution may facilitate easier identification of individual records. Ideally, an effective encoding method would produce a graph with fewer nodes and edges that are neither densely packed nor dispersed.

In our analysis, we employed the Titanic and NCVR datasets to evaluate the resistance of various encoding techniques—including bloom filter, two-step hash, table min-hash, and our encoding method—against graph-based re-identification attacks. The encoded datasets were visually transformed into graphs, allowing for an assessment of each method's defensive capabilities. Fig 6 shows the characteristics of each encoding method when the encoded data are graphically represented.

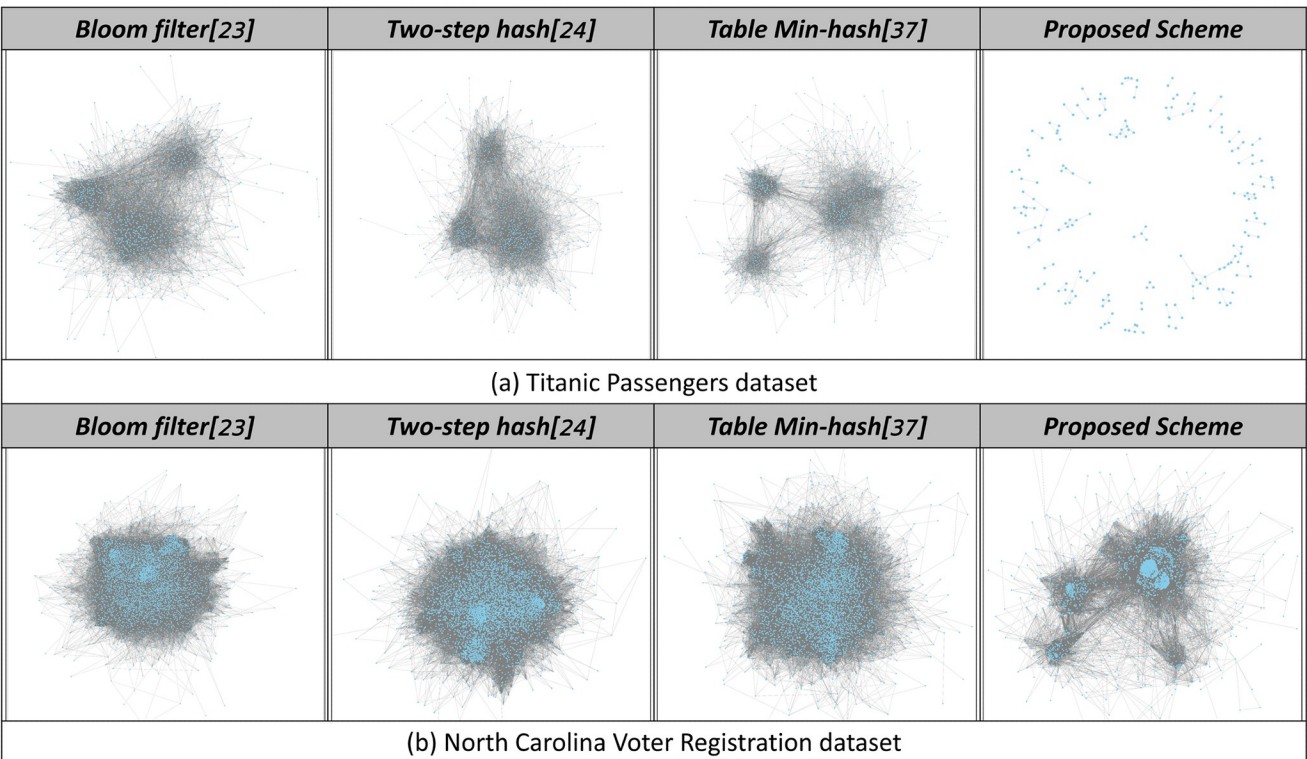

| Bloom filter[23] | Two-step hash[24] | Table Min-hash[37] | Proposed Scheme |
|---|---|---|---|
| (a) Titanic Passengers dataset | | | |
| Bloom filter[23] | Two-step hash[24] | Table Min-hash[37] | Proposed Scheme |
| (b) North Carolina Voter Registration dataset | | | |

**Fig 6. Graphical representation of the encoded dataset using each encoding method.**

- **Bloom Filter and Two-Step Hash:** These methods produced densely interconnected graphs for both the Titanic and NCVR datasets, revealing a fundamental susceptibility to re-identification attacks. The dense node connectivity in these graphs implies a higher risk of attribute similarity exploitation, which can lead to successful re-identification.

- **Tabulation Min-hash:** The graph for the Titanic dataset appeared slightly less dense compared to the previous methods, yet it still exhibited numerous extractable features that render it vulnerable to attacks. The NCVR dataset's graph, characterized by densely packed nodes and numerous edges, further confirmed the tabulation min-hash method's vulnerability to re-identification.

- **Zero-Relationship Encoding:** The zero-relationship encoding method demonstrated superior resistance to re-identification attacks. The Titanic dataset's graph, characterized by uniformly spread nodes with minimal edges, and the NCVR dataset's graph, with its nodes clustered into multiple groups and fewer edges, illustrate the method's effectiveness. The reduced node and edge density significantly complicates feature extraction and subsequent re-identification attempts, highlighting the zero-relationship encoding method's robustness.

In summary, the comparative analysis of graphical representations underscores the zero-relationship encoding method's enhanced defense against graph-based re-identification attacks. By minimizing node density and edge connectivity, our method substantially reduces the feasibility of exploiting attribute similarities, thereby preserving user privacy more effectively within PPRL frameworks.

**4.3.3. Threat model scenario.** PPRL systems facilitate the exchange of anonymized information among entities while safeguarding against unauthorized data breaches and re-identification threats. These systems, operating within a networked environment, are inherently exposed to a spectrum of security vulnerabilities, ranging from passive eavesdropping to active interference such as spoofing, sniffing, and snooping. The implications of such vulnerabilities extend to the unauthorized decryption and misuse of sensitive personal information, necessitating a robust defense mechanism inherent to the PPRL framework.

The design of the PPRL system is predicated on a semi-trust model among participating entities, who, while adhering to the established record linkage protocols, could potentially harbor honest-but-curious or outright malicious intents. The collaborative nature of the PPRL process involves the sharing of information necessary for record linkage, introducing a potential vector for the exploitation of anonymized data, thereby elevating the risk of re-identification without explicit consent from the data subjects.

Addressing the spectrum of threats, from honest-but-curious participants to external adversaries with malicious intent, the proposed zero-relationship encoding method employs a strategic limitation on the utilization of record attributes during the linkage process. This method leverages a selective subset of quasi-identifiers—attributes sufficiently generic yet vital for the linkage process such as name, sex, and age—thereby ensuring that the exposure of any single attribute does not compromise the integrity of the entire dataset. Moreover, the encoding process is further secured through the confidential agreement among participants on key parameters such as the hash function list and the attribute selection criteria, thereby obfuscating the encoding logic from potential adversaries.

The zero-relationship encoding approach fundamentally disrupts the direct correlation between the source and encoded records, thereby mitigating the risk associated with conventional encoding methodologies that preserve certain relational aspects susceptible to exploitation. By amalgamating multiple attributes to generate encoded values devoid of direct linkage to the original data, the method significantly impedes the adversaries' ability to apply cryptographic or pattern analysis techniques for data re-identification. This encoding paradigm, by obfuscating the relational mappings, effectively nullifies the utility of graph-based analysis tools in deciphering the encoded datasets, thereby fortifying the privacy assurances provided by the PPRL system.

In summary, the zero-relationship encoding method presents a comprehensive defense framework against the multifaceted threat landscape associated with PPRL systems. By ingeniously dissociating the encoded data from its source through a blend of attribute selection rigor and encoding complexity, the method not only thwarts attempts at unauthorized re-identification but also establishes a new benchmark for privacy preservation within the domain of PPRL.

## 5. Conclusion

In this study, we have introduced a zero-relationship encoding method to minimize the risk associated with record re-identification within PPRL systems. Our method significantly enhances privacy preservation by severing the link between source and encoded records, thus making reconstructing information from encoded data more challenging. The effectiveness of this approach was demonstrated through simulations of re-identification attacks, where our method exhibited superior resistance compared to existing encoding techniques. By considerably reducing the possibility of record re-identification, the proposed method ensures a higher degree of privacy protection in PPRL environments.

Future research should prioritize refining the proposed encoding technique for application in broader and more varied datasets. Additionally, further development is warranted to identify new measures that can further fortify privacy safeguards, contributing to a more robust and secure PPRL framework. We are committed to continuing our exploration into methods that enhance the security of private information. By focusing on these areas, future studies will be able to build upon our findings and extend the capabilities of privacy-preserving techniques in record linkage systems, ensuring a balance between privacy protection and the efficiency of linkage processes.

## Acknowledgments

The authors thank Anushka Vidanage for making his Python scripts of the graph matching attack available for this study.

## Author Contributions

**Conceptualization:** Sejong Lee, Yongseok Kwon.

**Data curation:** Sejong Lee, Yushin Kim.

**Formal analysis:** Sejong Lee.

**Methodology:** Sejong Lee.

**Resources:** Sejong Lee.

**Software:** Sejong Lee, Yushin Kim.

**Supervision:** Sunghyun Cho.

**Validation:** Sejong Lee.

**Visualization:** Sejong Lee.

**Writing – original draft:** Sejong Lee.

**Writing – review & editing:** Sejong Lee.

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
