## [Decision Letter · Decision Letter 0]

2 Feb 2024

PONE-D-23-43609Secure privacy-preserving record linkage (PPRL) system from re-identification attackPLOS ONE

Dear Dr. Lee,

Thank you for submitting your manuscript to PLOS ONE. After careful consideration, we feel that it has merit but does not fully meet PLOS ONE’s publication criteria as it currently stands. Therefore, we invite you to submit a revised version of the manuscript that addresses the points raised during the review process.

Kindly incorporate all the chnages recommended by all the reviewers and upload the revised manuscript by revision due date.

We look forward to receiving your revised manuscript.

Kind regards,

Radhakrishna Bhat

Academic Editor

PLOS ONE

‘This work was supported by Korea Research Institute for defense Technology planning 478 and advancement(KRIT) grant funded by the Korea government(DAPA(Defense 479

Acquisition Program Administration)) (No. KRIT-CT-22-021, Space Signal Intelligence 480 Research Laboratory, 2022)”

Please include this amended Role of Funder statement in your cover letter; we will change the online submission form on your behalf

“N/A”

Additional Editor Comments:

The reviewers have raised serious concerns on technical presentation and result comparison issues. Therefore, the article was peer reviewed with interest but has not been recommended for publication in its current form. I strongly recommend to reject this article and may be considerd after Major Revison.

Reviewers' comments:

Reviewer's Responses to Questions

**Comments to the Author**

1. Is the manuscript technically sound, and do the data support the conclusions?

Reviewer #1: No

Reviewer #2: Partly

Reviewer #3: Yes

2. Has the statistical analysis been performed appropriately and rigorously? 

Reviewer #1: Yes

Reviewer #2: N/A

Reviewer #3: Yes

3. Have the authors made all data underlying the findings in their manuscript fully available?

Reviewer #1: No

Reviewer #2: Yes

Reviewer #3: Yes

4. Is the manuscript presented in an intelligible fashion and written in standard English?

Reviewer #1: Yes

Reviewer #2: Yes

Reviewer #3: Yes

5. Review Comments to the Author

Reviewer #1: Secure privacy-preserving record linkage (PPRL) system from re-identification attack

Comments to authors:

1. The title of this paper, you must modify. Short form is not allowed in title.

2. Improve your abstract section.

3. Which are you used software for simulation. Show the simulation results.

4. What is the novelty of your work? Include the comparison table with published work.

5. Atleast you mention Figures in result section like through flowchart etc.

6. How are you justify the equation 1-2.

7. Encoding and decoding is old techniques, why are you using this?

8. Elaborate the result section with using of proper results. This paper is like a review paper.

Reviewer #2: 1. Please improve the Abstract.

2. The Introduction section is very poor. In a research article, the introduction section must be very strong with the motivations of this paper, which is missing in this paper. Moreover, the disadvantages of the existing schemes must be discussed to motivate this new work.

3. The point-wise contributions mentioned in the Introduction section are not specific.

4. The Related Work section is poor. The authors must include some more recent schemes. Also, the following papers must be cited to improve this section, as well as the Reference section:

a) Network anomaly detection using deep learning techniques

b) Problem-based cybersecurity lab with knowledge graph as guidance

c) Data accessing based on the popularity value for cloud computing

d) Local binary pattern-based reversible data hiding

e) Enhancing security of medical images using deep learning, chaotic map, and hash table

f) Review on offloading of vehicle edge computing

g) Security, privacy, trust, and anonymity

5. In section 2, a table can be given to summarize the entire section.

6. What is the use of the Zero-Relationship Encoding Method in the proposed scheme?

7. How the performance is increased?

8. In Eq. (3), how the value of n is decided?

9. Search stage is completely unclear.

10. Which entities are involved for key management?

11. How Eq. (5) improves the performance of the model?

12. The security analysis is not convincing at all.

13. What is the source of the dataset? Whether it is authentic or not? Mention clearly.

14. How the results of Figure 3 are generated?

15. Technical details about results are missing.

16. How the training is done?

17. What is the novelty of this work? It is hard to identify from the current version of this paper.

18. Key terms of the equations must be defined.

19. Use a well-known software to draw the diagrams of the results section.

20. The organization of the paper must be improved. The paper must be formatted properly.

21. Improve the English language.

22. The Reference section must be improved significantly.

23. Please give two paragraphs in the last section. One for concluding the entire chapter, and the second one for discussing future works. Also, complete it within 400 words.

24. Add section numbering

Reviewer #3: This manuscript discusses a security vulnerability in PPRL technology, introduces a proposed solution in the form of a zero-relationship encoding scheme, and provides simulation results and security analysis to demonstrate the effectiveness of the proposed method in preserving user privacy and resisting graph-based re-identification attacks. However, the simulation results and reidentification attacks should be justified with facts.

6. PLOS authors have the option to publish the peer review history of their article (what does this mean?). If published, this will include your full peer review and any attached files.

Reviewer #1: No

Reviewer #2: No

Reviewer #3: No

---

## [Author Response · Author response to Decision Letter 0]

20 Mar 2024

Dear Associate Editor and Reviewers,

Most of all, we would like to thank the associate editor and the reviewers for spending their valuable time and effort reviewing our paper. We have careful-ly read all of the reviewers' comments and have revised our manuscript ac-cording to reviewers' insightful comments and suggestions. Please find below our detailed replies to each of the comments.

Once again, we appreciate your kind and careful suggestions.

Sincerely,

Sunghyun Cho, Prof./Ph.D.

Dept. of Computer Science and Engineering.

Hanyang University ERICA

55 Hanyangdaehak-ro, Sangrok-gu, Ansan, Gyeonggi-do, Korea.

Tel: +82-31-400-5670 / Fax: +82-31-436-8152

Email: chopro@hanyang.ac.kr

---

## [Decision Letter · Decision Letter 1]

26 Aug 2024

PONE-D-23-43609R1Secure privacy-preserving record linkage system from re-identification attackPLOS ONE

Dear Dr. Lee,

Thank you for submitting your manuscript to PLOS ONE. After careful consideration, we feel that it has merit but does not fully meet PLOS ONE’s publication criteria as it currently stands. Therefore, we invite you to submit a revised version of the manuscript that addresses the points raised during the review process.

We look forward to receiving your revised manuscript.

Kind regards,

Zhiquan Liu, Ph.D.

Academic Editor

PLOS ONE

Journal Requirements:

Additional Editor Comments:

The author is requested to revise the manuscript with reference to the review comments, and then submit the manuscript as soon as possible for the next round of review.

Reviewers' comments:

Reviewer's Responses to Questions

**Comments to the Author**

1. If the authors have adequately addressed your comments raised in a previous round of review and you feel that this manuscript is now acceptable for publication, you may indicate that here to bypass the “Comments to the Author” section, enter your conflict of interest statement in the “Confidential to Editor” section, and submit your "Accept" recommendation.

Reviewer #1: All comments have been addressed

Reviewer #2: All comments have been addressed

Reviewer #4: (No Response)

2. Is the manuscript technically sound, and do the data support the conclusions?

Reviewer #1: Yes

Reviewer #2: Yes

Reviewer #4: Partly

3. Has the statistical analysis been performed appropriately and rigorously? 

Reviewer #1: Yes

Reviewer #2: Yes

Reviewer #4: (No Response)

4. Have the authors made all data underlying the findings in their manuscript fully available?

Reviewer #1: No

Reviewer #2: Yes

Reviewer #4: (No Response)

5. Is the manuscript presented in an intelligible fashion and written in standard English?

Reviewer #1: Yes

Reviewer #2: Yes

Reviewer #4: (No Response)

6. Review Comments to the Author

Reviewer #1: All figures have not good quality. You must improve the all figure quality. This is the basic need for any research paper.

Reviewer #2: The authors have addressed all the previous comments. No further comments to address. This paper can be accepted.

Reviewer #4: This study proposes a zero-relationship encoding scheme that minimizes the linkage between source and encoded records to enhance PPRL systems' resistance to re-identification attacks. The study is good overall, but there are some flaws.

The comparison in Table 1 is too general. Please list the performance metrics considered in this article, and then analyze whether each method supports these metrics to make it more intuitive. The numbering of some formulas is missing and needs to be carefully modified. Some important literature discussions are missing, such as efficient privacy-preserving spatial range query over outsourced encrypted data, efficient privacy-preserving spatial data query in cloud computing, which suggests the author to supplement the discussion of these literatures. It is suggested that the author add some experiments to prove the advanced nature of the proposed scheme.

7. PLOS authors have the option to publish the peer review history of their article (what does this mean?). If published, this will include your full peer review and any attached files.

Reviewer #1: No

Reviewer #2: No

Reviewer #4: No

---

## [Author Response · Author response to Decision Letter 1]

17 Sep 2024

we have made revisions to the manuscript based on the reviewers' comments, and we have addressed all their suggestions. The revised manuscript and a detailed response to the reviewers’ comments are attached for your review.

---

## [Decision Letter · Decision Letter 2]

1 Nov 2024

PONE-D-23-43609R2Secure privacy-preserving record linkage system from re-identification attackPLOS ONE

Dear Dr. Lee,

Thank you for submitting your manuscript to PLOS ONE. After careful consideration, we feel that it has merit but does not fully meet PLOS ONE’s publication criteria as it currently stands. Therefore, we invite you to submit a revised version of the manuscript that addresses the points raised during the review process.

We look forward to receiving your revised manuscript.

Kind regards,

Zhiquan Liu, Ph.D.

Academic Editor

PLOS ONE

Journal Requirements:

Additional Editor Comments:

The reviewers partially affirmed the contribution of this paper and put forward suggestions for modification. The authors are requested to revise as much as possible and submit the revised version. The final acceptance depends on the opinions of the next round of reviewers.

Reviewers' comments:

Reviewer's Responses to Questions

**Comments to the Author**

1. If the authors have adequately addressed your comments raised in a previous round of review and you feel that this manuscript is now acceptable for publication, you may indicate that here to bypass the “Comments to the Author” section, enter your conflict of interest statement in the “Confidential to Editor” section, and submit your "Accept" recommendation.

Reviewer #2: All comments have been addressed

Reviewer #4: (No Response)

Reviewer #5: All comments have been addressed

2. Is the manuscript technically sound, and do the data support the conclusions?

Reviewer #2: Yes

Reviewer #4: (No Response)

Reviewer #5: Yes

3. Has the statistical analysis been performed appropriately and rigorously? 

Reviewer #2: N/A

Reviewer #4: (No Response)

Reviewer #5: Yes

4. Have the authors made all data underlying the findings in their manuscript fully available?

Reviewer #2: Yes

Reviewer #4: (No Response)

Reviewer #5: Yes

5. Is the manuscript presented in an intelligible fashion and written in standard English?

Reviewer #2: Yes

Reviewer #4: (No Response)

Reviewer #5: Yes

6. Review Comments to the Author

Reviewer #2: The authors have addressed all the previously raised comments. Thus, this paper can be considered for publication.

Reviewer #4: It seems that the author did not revise the paper carefully according to the previous opinions, and it is suggested that the author should revise the paper carefully before being considered for employment.

Reviewer #5: The author has incorporated all suggestions in the paper:

Originality of paper is now looking much better.

Technical merit of paper is now looking good.

The overall manuscript have a good qualitative work.

7. PLOS authors have the option to publish the peer review history of their article (what does this mean?). If published, this will include your full peer review and any attached files.

Reviewer #2: No

Reviewer #4: No

Reviewer #5: No

---

## [Author Response · Author response to Decision Letter 2]

1 Nov 2024

Thank you for your continued guidance regarding our manuscript. We appreciate your attention to detail in ensuring the highest standards.

We would like to address the concern about retracted references thoroughly. In preparing this revision, we carefully re-evaluated the 50 cited articles and 1 open dataset to ensure their validity and accessibility. We verified each reference individually, using Google Scholar, PubMed, CrossRef, and visiting each article’s online source to check for any retraction status. Through this process, we confirmed that none of the references in our paper have been retracted, and all remain accessible.

The question regarding retracted articles was previously raised during an earlier review, and at that time, we conducted a thorough check with similar due diligence. We found no issues then, and this subsequent evaluation also confirmed there were no problems. If there is a specific reference that has raised concerns, we would greatly appreciate clarification so we can address it directly. This current round of review marks our second verification of the references, and both checks have yielded no problematic findings.

Regarding the revisions based on reviewer comments, we have worked diligently in each revision round to incorporate all feedback fully and thoughtfully. These responses were detailed in our reply letters. Throughout this process, our manuscript has been reviewed by two distinct sets of reviewers due to a change in editors. We have endeavored to balance these varied perspectives carefully, ensuring all feedback was addressed without conflicts.

Thank you again for the opportunity to clarify, and please do let us know if there are specific references that require further attention.

Kind regards,

Sejong Lee

---

## [Editor Report · Decision Letter 3]

12 Nov 2024

Secure privacy-preserving record linkage system from re-identification attack

PONE-D-23-43609R3

Dear Dr. Lee,

We’re pleased to inform you that your manuscript has been judged scientifically suitable for publication and will be formally accepted for publication once it meets all outstanding technical requirements.

Kind regards,

Prof. Zhiquan Liu

Academic Editor

Institution: Jinan University

Email: zqliu@vip.qq.com

WeChat: 1565315

Homepage: https://www.zqliu.com

Welcome to contact me for cooperation.

Additional Editor Comments (optional):

Accept
---

## [Editor Report · Acceptance letter]

13 Nov 2024

PONE-D-23-43609R3 

PLOS ONE

Dear Dr. Lee, 

I'm pleased to inform you that your manuscript has been deemed suitable for publication in PLOS ONE. Congratulations! Your manuscript is now being handed over to our production team.

Kind regards, 

on behalf of

Professor Zhiquan Liu 

Academic Editor

PLOS ONE